# Prenatal Alcohol Exposure in Rats Diminishes Postnatal Cxcl16 Chemokine Ligand Brain Expression

**DOI:** 10.3390/brainsci10120987

**Published:** 2020-12-15

**Authors:** Pedro Juárez-Rodríguez, Marisol Godínez-Rubí, Carolina Guzmán-Brambila, Edgar Padilla-Velarde, Arturo Orozco-Barocio, Daniel Ortuño-Sahagún, Argelia E. Rojas-Mayorquín

**Affiliations:** 1Departamento de Ciencias Ambientales, Universidad de Guadalajara, Centro Universitario de Ciencias Biológicas y Agropecuarias, Guadalajara 45200, Mexico; psic.pjr@gmail.com (P.J.-R.); edgarpadilla0@outlook.es (E.P.-V.); 2Laboratorio de Neuroinmunobiología Molecular, Instituto de Investigación en Ciencias Biomédicas (IICB), Departamento de Biología Molecular y Genómica, Universidad de Guadalajara, Centro Universitario de Ciencias de la Salud, Guadalajara 44340, Mexico; 3Laboratorio de Investigación en Patología, Departamento de Microbiología y Patología, Universidad de Guadalajara, Centro Universitario de Ciencias de la Salud, Guadalajara 44340, Mexico; juliana.godinez@academicos.udg.mx; 4Tecnologico de Monterrey, Escuela de Medicina, Campus Guadalajara, Zapopan 45201, Mexico; caro@tec.mx; 5Laboratorio de inmunobiología, Departamento de Biología Celular y Molecular, Universidad de Guadalajara, Centro Universitario de Ciencias Biológicas y Agropecuarias, Guadalajara 45201, Mexico; arorozcob@prodigy.net.mx

**Keywords:** prenatal alcohol exposure, neurodevelopmental disorders, Cxcl16, microarrays

## Abstract

Maternal ethanol consumption during pregnancy is one of the main causes of Neurodevelopmental disorders (NDD). Prenatal alcohol exposure (PAE) produces several adverse manifestations. Even low or moderate intake has been associated with long-lasting behavioral and cognitive impairment in offspring. In this study we examined the gene expression profile in the rat nucleus accumbens using microarrays, comparing animals exposed prenatally to ethanol and controls. Microarray gene expression showed an overall downward regulatory effect of PAE. Gene cluster analysis reveals that the gene groups most affected are related to transcription regulation, transcription factors and homeobox genes. We focus on the expression of the C-X-C motif chemokine ligand 16 (*Cxcl16*) which was differentially expressed. There is a significant reduction in the expression of this chemokine throughout the brain under PAE conditions, evidenced here by quantitative polymerase chain reaction qPCR and immunohistochemistry. Chemokines are involved in neuroprotection and implicated in alcohol-induced brain damage and neuroinflammation in the developing central nervous system (CNS), therefore, the significance of the overall decrease in Cxcl16 expression in the brain as a consequence of PAE may reflect a reduced ability in neuroprotection against subsequent conditions, such as excitotoxic damage, inflammatory processes or even hypoxic-ischemic insult.

## 1. Introduction

Maternal ethanol consumption during pregnancy is one of the main causes of Neurodevelopmental disorders (NDD) [1], neuropsychiatric diseases that generate disability in social, motor, cognitive, affective and language areas [2]. Even low or moderate intake has been associated with long-lasting behavioral and cognitive impairment in offspring [3,4,5]. Prenatal alcohol exposure (PAE) produces several adverse manifestations that range from relative normality to perinatal mortality, encompassed under the term fetal alcohol spectrum disorders (FASD) [6]. Moreover, PAE exacerbates the activation of mesolimbic pathways affecting vulnerable regions related to conditioned or motivated behavior [7] and regulation of impulsivity [8], hallmark features of some NDDs.

PAE has been targeted as a risk factor for different NDDs. FASD diagnosis is associated with a higher risk of Attention Deficit Hyperactivity Disorder (ADHD) and, reciprocally, ADHD diagnosis is a highly probable antecedent of FASD [9]. Furthermore, parallelism between FASD and ADHD has been established [10,11], highlighting that there may be common etiological pathways.

In this study, we examined the gene expression profile in the rat nucleus accumbens using microarrays, comparing animals exposed prenatally to ethanol and controls. We focused on the expression of the C-X-C motif chemokine ligand 16 (*Cxcl16*), which was differentially expressed under the experimental conditions identified in the microarray analysis and confirmed by qPCR. Chemokines are known largely for their role in the immune system, but chemokine signaling is also involved in the regulation of neural cell proliferation, migration and survival during neurodevelopment [12], in synaptic transmission [13] and in the maintenance of homeostasis in the adult CNS modulating neuroprotective processes [14] against excitotoxic damage and hypoxic insult [15]. Interestingly, mutations in chemokine signaling genes have been detected in patients with NNDs [16]. Moreover, the chemokine signalling may mediate alcohol-induced brain damaged and neuroinflammation in the developing CNS [17], suggesting the possible role of an aberrant chemokine signalling in these disorders.

There is currently insufficient understanding of the underlying molecular mechanisms of the cognitive, affective, and behavioral deficits elicited by PAE which partly characterize other NDDs, highlighting the need for wider experimental approaches. Therefore, this study aims to uncover *Cxcl16* as a new possible relevant player in rat brains following PAE.

## 2. Materials and Methods

### 2.1. Experimental Animals

Wistar rats (Charles River) were delivered on postnatal day 90 and maintained in groups of three to five per cage at a controlled temperature (23 ± 1 °C), on a 12-h light/dark cycle (lights on 7:00 a.m.) with free access to food and water. Two groups of rats were used, one for the microarray analysis and qPCR: the other for Immunohistochemistry. All the animal procedures were carried out in compliance with the Mexican Official Norm: NOM-062-ZOO-1999, and the guidelines of the local institutional animal care and use committee. All efforts were made to minimize the number of animals used and their suffering. This project was approved by the Ethical Committee of the Instituto de Neurociencias in the Centro Universitario de Ciencias Biológicas y Agropecuarias, Universidad de Guadalajara (ET-112016-225).

### 2.2. Prenatal Alcohol Treatment

The prenatal alcohol administration protocol used in this study is described elsewhere [18,19]. Pregnant Wistar rats weighing 250–300 g at treatment onset were treated with ethanol (EtOH) (20% *w/v* in a 0.9% saline solution) by intragastric gavage between gestational days (GD) 8–20, to mimic the drinking behavior that produces high blood ethanol concentrations related with increased risk of FASD in humans. Intragastric gavage allows for precise control of the dose of alcohol administered. The daily dose was 6.0 g/kg, divided into two 3.0 g/kg doses, administered 5–6 h apart on weekdays. A single daily dose of 4 g/kg ethanol was given on weekends. Blood ethanol concentration was measured in a previous study 1.5 h after the second daily dose of ethanol, and was between 281 and 341 mg/dL on GD 20 [20]. The control group received a solution of sucrose (ISO) 10.5 g/kg (in 0.9% saline) divided into two 5.25 g/kg doses to substitute for ethanol isocalorically on weekdays. A single daily dose of 7.0 g/kg was administered on weekends. This isocaloric group has been used as a control group in previous studies using this model, showing almost no interference [21]. In addition, although the relationship between the reward circuit and the orexigenic peptides have recently been documented in the nucleus accumbens [22], nevertheless in studies of PAE, the isocaloric group and the untreated group show no differences in the levels of expression of these peptides [23,24].

On postnatal day (PND) 1, pups were individually examined for gross physical abnormalities. Each experimental group included 10 male pups from the ISO group and 10 male pups from the EtOH group. Pups were maintained in each litter until weaning. At weaning, male pups were separated and placed in collective cages in groups of 4–5. Female pups were sacrificed by decapitation according to the NOM-062-ZOO-1999 in correspondence with the Euthanasia section of the Guide for the Care and Use of Laboratory Animals. Only male offspring were used, given the higher prevalence of ADHD in human males with respect to females [25].

### 2.3. Sample Preparation

Thirty-eight-day old pups were used in the present study, corresponding to human adolescence [26]. Microarray analysis was performed on pools of seven rats for each group (i.e., seven ISO, seven EtOH) to compensate for inter-organism variability. Six rats were used for qPCR, and three for immunohistochemistry from each group.

### 2.4. Extraction, Quantification, and Differential Expression of mRNA in Microarrays

Total RNA was isolated from the nucleus accumbens of seven prenatally EtOH exposed pups and seven ISO as previously described [27]. Briefly, tissue was homogenized in the presence of the TRIzol reagent (Invitrogen, Carlsbad, CA, USA) and after chloroform was added, RNA was precipitated from the aqueous phase with isopropanol at 4 °C. The RNA was reconstituted in RNase-free water, and then quantified in a NanoDrop™ OneC microvolume spectrophotometer (Thermo Fisher Scientific, Wilmington, DE, USA) using 1 μL of the sample in RNase-free water. Two RNA pools were then created per group, each with equal amounts of RNA from the seven rats. The two RNA samples were analyzed in two independent arrays.

### 2.5. Printing, Probe Preparation, Hybridization, Data Acquisition and Analysis of the Array

A *Mus musculus* 22,000 65-mer Oligo Library from a Sigma-Genosys set (The Woodlands, TX, USA) was probed with the RNA isolated from the whole nucleus accumbens in a heterologous hybridization [28]. Complementary DNA (cDNA) was synthesized from 10 μg of total RNA as the template, incorporating dUTP-Cy3 or dUTP-Cy5, and equal quantities of the labeled cDNAs were hybridized to the 22,000 oligo mouse arrays, as previously described [29,30]. The array images were acquired and quantified in a ScanArray 4000 apparatus using the accompanying ScanArray 4000 software (Packard BioChips Technologies; Billerica, MA, USA). Finally, microarray data were analyzed using genArise v2.0 [31] free software to identify genes that are good candidates for differential expression by calculating an intensity-dependent Z-score. This software uses a sliding window algorithm to calculate mean and standard deviation within a window surrounding each data point, and defines a Z-score where Z measures the number of standard deviations a data point is from the mean. According to the genArise analysis, the genes with a Z-score ±1.5 are considered statistically significant in difference in expression (*p* < 0.05). However, to be more rigorous, we select only those genes with z-score over ±2.0 (a higher selective Z-score), to be included for further analysis in the online Database for Annotation, Visualization, and Integrated Discovery (DAVID) Bioinformatics Resources v.6.8. (see the next section [29,32,33]. The microarray data were deposited in the GEO database, which is in line with the MIAME (Minimum Information About a Microarray Experiment) and MINSEQE (Minimum Information About a Next-generation Sequencing Experiment) guidelines.

### 2.6. Functional Classification of Differentially Expressed Genes

The list containing differentially expressed genes in the microarray was generated with genArise software based on a cutoff Z-score of ±2.00. To assess genes and processes in the response to EtOH the list was analyzed using the Functional Annotation Clustering available through the DAVID Bioinformatics Resources v.6.8. [34,35], which provides a rapid means of organizing large lists of genes into functionally related groups.

### 2.7. qPCR mRNA Quantification

Total RNA was isolated from nucleus accumbens and prefrontal cortex tissue, using the TRIzol reagent (following the manufacturer’s instructions), quantified at 260 nm and RNA purity was determined from the A260/280 ratio in a NanoDrop™ OneC microvolume spectrophotometer using 1 μL of the sample in RNase-free water. Reverse transcription-Polymerase chain reaction (RT-PCR) was performed with a total concentration of 1 µg of the messenger RNA (mRNA) using the iScript^TM^ cDNA Synthesis Kit (Bio-Rad Laboratories, Hercules, CA, USA) with the thermal cycler ProFlex PCR System (Applied Biosystems, Foster City, CA, USA). Subsequently, qPCR was performed in 96-well plates with the StepOnePlus Real-time PCR System (Roche, Basel, Switzerland), using TaqMan probes (Applied Biosystems) for *Fkbp5*, *Cxcl16*, *Hmg20b*, and *Hoxb13* following the manufacturer’s instructions. These genes were selected based on the functional annotation analysis performed. Reactions were run following a standard ramp speed protocol using 12 μL volumes. PCR cycling consisted of a 2 min initiation at 50 °C, followed by 40 cycles consisting of a 20 s denaturation at 95 °C and an anneal and extension at 60 °C for 30 s. All experiments included six biological replicates per treatment and two technical replicates per sample. The data were analyzed utilizing the comparative cycle threshold (Ct) method (ΔΔCt) [36] with glyceraldehyde three phosphate dehydrogenase (*Gapdh*) gene used as constitutive control for normalization. According to the comparative Ct method, PCR data are reported as fold change values.

### 2.8. Immunohistochemistry

Rats were anesthetized by peritoneal injection of pentobarbital (50 mg/kg) and sacrificed by decapitation. Immediately after, brains were rapidly dissected and sectioned into coronal 50-μm-thick slices with a stainless-steel matrix (RWD Life Science 68711, Dover, DE, USA) fitted for rat brains. The use of the matrix allows a more homogeneous comparison of the sections, with the advantage of being a simple, reproducible method that significantly reduces tissue handling. Then the tissues were fixed for 24 h in 10% neutral buffered formalin (NBF), embedded in paraffin for preservation and cut into 3 μm sections for mounting on pre-loaded slides. Tissue-sections were routinely processed by heat and re-hydration in a xylol-alcohol series. Once tissue sections were re-hydrated, heat-induced epitope retrieval was accomplished in a bath of sodium citrate solution 10 mM (pH = 6). Endogenous peroxidase activity was neutralized with H_2_O_2_ 3% for 10 min. The tissue sections were then incubated for 2 h at room temperature with the primary antibody anti-CXCL16 (cat. sc-514363; dilution 1:50; Santa Cruz Biotechnology, Inc., Dallas, TX, USA). The detection of primary antibody was performed with HiDef Detection^TM^ HRP Polymer System (cat: 954D; Cell Marque, Rocklin, CA, USA) following the manufacturer’s instructions. Sections were further incubated with the substrate/chromogen, 3,3′-diaminobenzidine (DAB) for 10 min and were then counterstained with hematoxylin, dehydrated and mounted. Images of the tissues were captured with a digital camera (Axiocam ICc 1305; Zeiss AG, Oberkochen, Germany) attached to an optical microscope (Axio Lab.A1; Zeiss AG). For validation and standardization of Cxcl16 antibody staining, we performed immunohistochemistry in positive external anatomical controls tissues with known constitutively expression of Cxcl16 (both from human and rat). Results are presented in Appendix A.

### 2.9. Data Analysis

The results are expressed as the mean ± standard error of the mean (SEM). Statistical analysis was performed with GraphPad Prism v.6 software (San Diego, CA, USA) applying the Student *t*-test. Values < 0.05 were considered statistically significant.

## 3. Results

### 3.1. Changes in Patterns of Gene Expression in the Nucleus Accumbens after PAE

We first assessed the overall pattern of gene expression in the microarray. Using the geneArise software we set a cut-off Z-score of ±2.00 to generate a reduced gene list to continue with downstream analysis. In the resulting ISO–EtOH gene list there were 945 differentially expressed genes, of which 636 genes decreased their expression and only 309 genes increased. The pattern of gene expression showed a general downregulating effect for EtOH. Complete lists of genes are included in Appendix A. Obtained results from microarray hybridization were entered in the GEO database under the accession number: GSE160433.

### 3.2. Functional Classification Analysis of Genes Affected by PAE

We searched for groups of genes based on the functional similarity of genes in the list by using DAVID v.6.8. Functional Annotation Clustering setting. For clustering stringency, we set the Kappa similarity term overlap to 3 and Kappa similarity threshold to 0.85. We used an enrichment score > 1.5 as a criterion for notable enrichment [34]. Results indicating the most significant nonredundant “functions annotations” for upregulated and downregulated genes are shown in Table 1.

### 3.3. qPCR Validation of Microarray Results

Considering the gene expression profile generated by the analysis of microarrays, and to further confirm the expression changes of some genes, we selected four genes with high z-scores and that were representative of functions of interest. The selected genes were *Hmg20b* (Z = −3.96), *Hoxb13* (Z = −3.36), *Cxcl16* (Z = −2.36), and *Fkbp5* (Z = −2.35). For ISO–EtOH, the downregulation of FK506 binding protein 5 (Fkbp5) (*p* = 0.02) and C-X-C motif chemokine ligand 16 (Cxcl16) (*p* = 0.04) was confirmed, under these experimental conditions (Figure 1). The high mobility group 20B (Hmg20b) (*p* = 0.29) was not confirmed. Because there are commercially available antibodies for Cxcl16 and Fkbp5, we further explored their brain expression by immunohistochemistry. For Hmg20b there is no antibody available, so further analysis of its expression is not currently possible. Finally, for the confirmation of homeobox B13 (Hoxb13) by real-time PCR, we obtained a commercially available probe that only corresponded to the mouse gene sequence (the rat sequence was not available). Consequently, we were unable to amplify this gene for verification. Further analysis needs to be undertaken in this case.

Additionally, we determined the expression of the selected genes on the prefrontal cortex, since this region also plays key roles in mesolimbic pathways related to the modulation of conditioned or motivated behavior [37,38] and regulation of impulsivity [39]. *Fkbp5* showed an increase in its expression (*p* = 0.01) opposite to our findings in the nucleus accumbens. *Cxcl16* showed only a tendency to be downregulated, although this was not significant (*p* = 0.17) and *Hmg20b* did not show significant changes in its relative expression (*p* = 0.30) under these experimental conditions (Figure 2).

### 3.4. Immunohistochemistry

Because the expression of Fkbp51 under alcohol exposure has been previously described [40], we then proceeded to analyze the expression of Cxcl16 protein by immunohistochemistry. Cxcl16 was expressed in the cytoplasm of neurons and, to a lesser extent, in glial cells. The expression of Cxcl16 showed a regionalized differential expression that was particularly remarkable in the nucleus accumbens, prefrontal cortex, hippocampus, striatum and periaqueductal gray substance. Other regions where expression was observed were the medial longitudinal fasciculus and the motor and somatosensory cortex. In general, the expression was of greater intensity and in a greater number of cells in the ISO group. In the nucleus accumbens, an intense expression was found in the shell region in pyramidal and polygonal soma neurons in the ISO group. Adjacent to this zone, the granular neurons of the islets of Calleja (a structure of the limbic system and part of the reward system), exhibited cytoplasmic staining apparently of moderate intensity. In the EtOH group, the somas and neuropile of the shell region also showed a suggestively moderate staining (Figure 3). Quantitative methods are required to complement the results to confirm our observations.

In the prefrontal cortex, at the level of the cingulate cortex, the expression of Cxcl16 in the ISO group ranged from moderate to high intensity varying between the different layers, and was found in all layers of the cortex, both in the neuronal somas and in the neuropile. Under PAE, by contrast, the intensity of the expression was slight, and was concentrated in the cytoplasm of few pyramidal neurons of the III layer (Figure 4).

The pyramidal neurons of the CA1 sector of the hippocampus, in the ISO group, displayed a moderate intensity staining, while in the EtOH group, no significant expression was observed. In the striatum, both groups exhibited expression in the somas of the neurons arranged between the slender fascicles of the white matter, although the expression in the ISO group was of greater magnitude and extended to the neuropile compared to the EtOH group. In the periaqueductal grey substance, expression of Cxcl16 was identified in both groups, albeit intensely and occurring in a greater number of neurons in the ISO group (Figure 5).

## 4. Discussion

PAE is one of the main causes of NDDs, and has been related to FASD [41], ADHD [9,10,11], mental disability and autism spectrum disorder [42,43], although the underlying molecular mechanism potentially common to various alcohol-related NDDs remains unclear. Moreover, the most common methods for studying the effects of EtOH on neurodevelopment rely on in vitro systems whose results are difficult to extrapolate due to the lack of the environment provided by the maternal uterus and the placenta, or the acute in vivo administration that fails to faithfully reproduce the range of manifestations caused by continued exposure to EtOH during pregnancy. In this study, we used an in vivo model of chronic prenatal exposure to EtOH that has shown good results mimicking the main characteristics of FASD [11]. We set out to assess the effects of prenatal EtOH exposure on gene expression in rats.

Most gene expression studies are performed with a “candidate gene” approach to evaluate the effects of PAE by screening for specific genes. However, in order to tackle the complexity of disorders caused by maternal alcohol consumption, it is necessary to resort to broader approaches that allow us to improve our understanding of the molecular basis underlying the alterations that last until adulthood. Results of the initial microarray analysis identifying the effects of PAE (ISO vs. EtOH) are consistent with previous microarray studies indicating that EtOH has a general suppressive effect on gene expression [43,44,45,46]. Nevertheless, the study by Mandal et al. had a greater number of overexpressed genes (194) compared to those that reduced their expression (104) [47], which could be due to the spatial and temporal changes in cues that guide gene expression in different periods of neurodevelopment, since the microarray was hybridized with embryonic (E18) hippocampal tissue, compared to most microarrays made with tissue from youth or adult organisms.

PAE adverse effects may involve changes in fetal programming, a response to non-genetic and environmental factors that permanently organizes physiological and neurobiological systems, though epigenetic mechanisms [48]. In this study, we focused on a few selected genes modified by PAE. First, *Fkbp5* encodes a chaperone protein involved in the negative feedback of the HPA axis, forming protein complexes with Hsp90 and the glucocorticoid receptor, which retains the latter in the cytoplasm, therefore, preventing it from translocating to the nucleus [49]. Caldwell et al. reported a diminished expression of *Fkbp5* in PAE rat hippocampi [40], conversely, McClintick et al. found it overexpressed in hippocampi of alcohol use disorder patients in a post-mortem study [50]. We found *Fkbp5* to be under-expressed in nucleus accumbens, but over-expressed in the prefrontal cortex, which could mean that the effects of PAE on the expression of this gene are heterogeneous throughout the whole brain, even if the entire organ is exposed to the same doses [51]. To understand its effects on this gene expression further studies are required.

On the other hand, the *CXCL16* gene encodes a transmembrane chemokine produced mainly by brain endothelial cells and by reactive astrocytes, which can upon cleavage by metalloproteinases (ADAM 10, 17) be released into the extracellular space. Its receptor the CXCR6 is expressed in astrocyte precursor cells [52]. In the immunohistochemical study, we documented the basal expression of CXCL16 in the control group, which was evident in the cingulate cortex, nucleus accumbens, hippocampus, striatum and periaqueductal gray substance. These anatomical regions are known to be part of the cortico-striatal-limbic circuit which participates in the response to substance consumption and in the reward system [53], so it is relevant that CXCL16 showed changes in its expression when exposed to ethanol in these regions. The effect of ethanol exposure was variable and regionalized, with a tendency to decrease protein expression in all the mentioned regions. This effect is congruent with the decrease in gene expression observed in the nucleus accumbens and in the prefrontal cortex. Interestingly, there was no brain region that overexpressed CXCL16 with ethanol exposure, which seems to indicate a common regulation mechanism in all these regions.

Although CXCL16 is considered a proinflammatory chemokine, its role as a neuroprotective molecule has been studied in vitro under conditions of excitotoxicity [54] and hypoxia [15], when interacting with astrocytes and microglia. In addition, its ability to modulate excitatory and inhibitory activity in the CA1 region of the hippocampus has been documented. This effect seems to involve adenosine and at least one of its receptors (A3R) [13,15,54]. On the other hand, it is widely reported that ethanol modifies the levels of adenosine and its receptors in acute and chronic exposure [55], so this pathway could be a common point between ethanol and CXCL16.

Another possible interaction between CXCL16 and ethanol is through the PI3K-AKT pathways. Its activity is involved in the activation of the PI3K-AKT-ERK pathway during neurodevelopment to induce glial cell proliferation and migration, and in response to damage favoring astrogliosis [12]. It is also documented that under hypoxic conditions, CXCL16 increases the phosphorylation of target molecules in the PI3K-AKT-GSK3β pathway [56]. Additionally, chronic alcohol exposure and neuroadaptation states induced in the prefrontal cortex have been related to overactivation of this same pathway [57].

Furthermore, it has been reported that CXCL16 intervenes in modulating GABAergic and Glutamatergic activity in the CA1 area of the hippocampus [13]. It is worth mentioning that some *CXCL16* polymorphisms have recently been associated with schizophrenia [58]. We found *Cxcl16* to be under-expressed in nucleus accumbens by the effect of PAE, which suggests that it might be contributing to a GABA/Glutamate imbalance caused by the changes in the levels of this chemokine. Other chemokines, such as CXCL12 [59] and CCL2 [17,23,24] have been associated with PAE. Chang et al. reported the increased expression of Cxcl12-Cxcr4 by PAE [59]. In contrast, in the present study, we describe, for the first time, that PAE downregulates Cxcl16 expression in different regions of the brain, highlighting the intricate roles of chemokine systems and the temporal- and regional-dependent vulnerability of the developing brain to the effects of alcohol.

Lepore et al. demonstrated that in physiological conditions microglial cells respond to CXCL16 stimulation developing an anti-inflammatory phenotype and, in the context of inflammation (LPS-IFNγ), CXCL16 attenuates the pro-inflammatory microglia phenotype [60]. Supporting the hypothesis that CXCL16 release is an attempt to limit neuronal damage, a chemokine signalling cascade to counteract ischemic brain damage has recently been described [14] by which the astrocyte release of CXCL12 promotes neuronal shedding of CXCL3 and ADAM17, these molecules, in turn, promoting the release of CXCL16 from glia. CXCL16 overexpression, alongside the activation of adenosine type 3 receptor (A3R), induces the release of CCL2 from astrocytes, an important molecule for mediating neuroprotection activating pro-survival pathways in neurons. The downregulation of Cxcl16 in the present study suggests that the neuroprotective mechanism orchestrated by several chemokines and cellular types is impaired by PAE in the brain regions studied.

## Figures and Tables

**Figure 1 brainsci-10-00987-f001:**
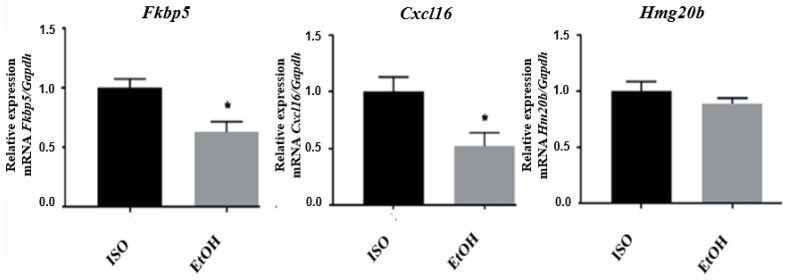
mRNA quantification and relative expression of genes altered by EtOH exposure in the nucleus accumbens. Fold change < 1 indicates reduced expression in response to ethanol relative to a value of 1. * *p* < 0.05.

**Figure 2 brainsci-10-00987-f002:**
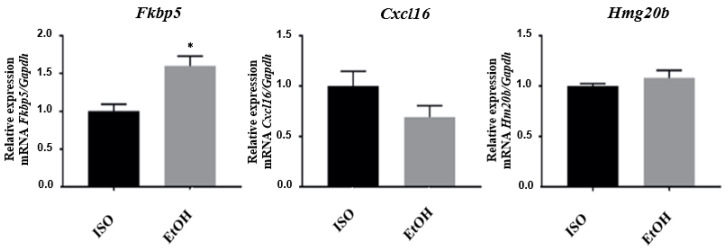
mRNA quantification and relative expression of genes altered by EtOH exposure in the prefrontal cortex. Fold change <1 indicates reduced expression in response to ethanol relative to a value of 1.* *p* < 0.05.

**Figure 3 brainsci-10-00987-f003:**
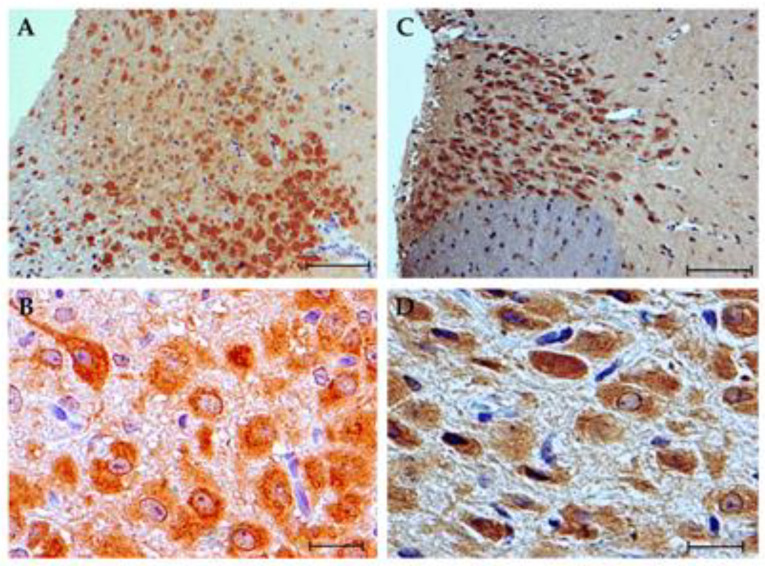
Cxcl16 expression in nucleus accumbens. (**A**,**B**) ISO group. An intense and diffuse expression is observed in the soma of numerous neurons in the shell zone of the nucleus accumbens which contrasts in this case with the negativity of the glial cells. (**C**,**D**) EtOH group. Cytoplasmic expression appreciated in abundant cells of slightly lower intensity vs. the ISO group. Scale bar in (**A**,**C**) = 100 μm. Scale bar in (**B**,**D**) = 20 μm.

**Figure 4 brainsci-10-00987-f004:**
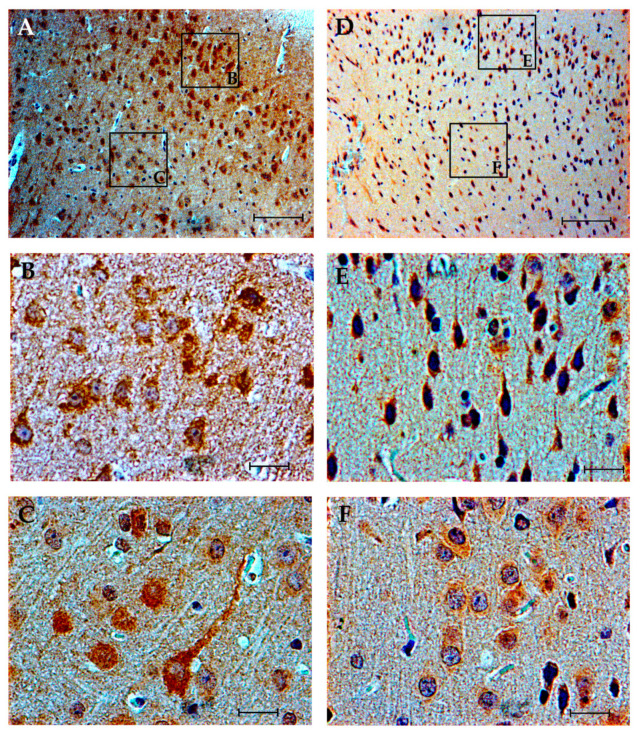
Cxcl16 expression in cingulate cortex (cg1 and cg2). (**A**–**C**) ISO group. Cytoplasmic expression appreciated in the soma of neurons of all layers, with predominance in zone III and V, as well as in the neuropile. (**D**–**F**) EtOH group. Expression persists in the neuronal soma; however, the intensity of expression is notably lower and seems to be restricted to layer V, with few positive cells in layer III. In the neuropile, the intensity of expression is significantly lower. Scale bar in (**A**,**D**) = 100 µm. Scale bar in (**B**–**D**,**F**) = 20 µm.

**Figure 5 brainsci-10-00987-f005:**
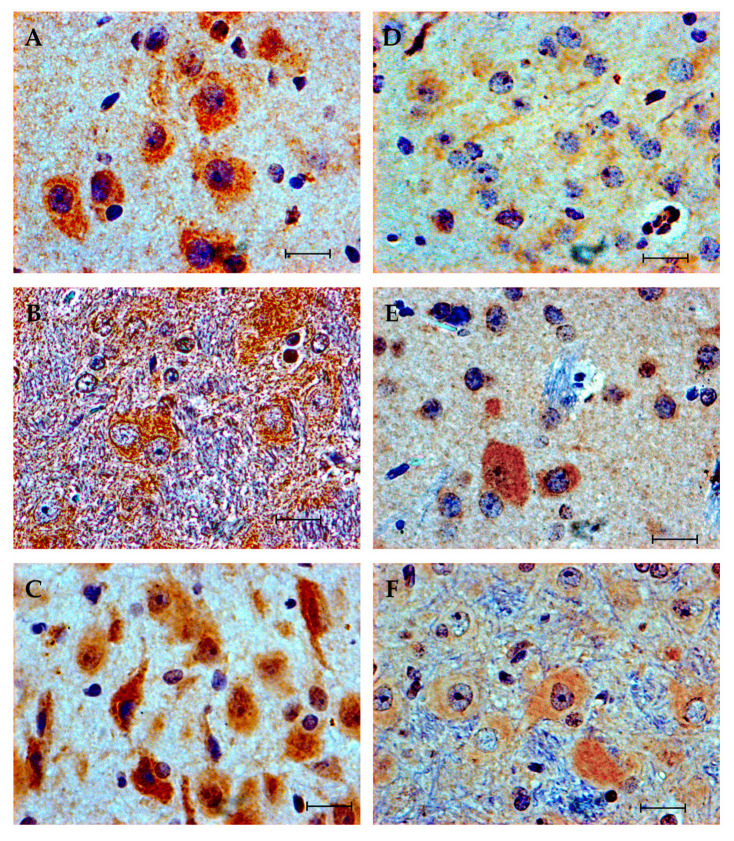
Cxcl16 expression in the hippocampus (**A**,**D**), striatum (**B**,**E**) and periaqueductal gray substance (**C**,**F**). ISO group (**A**–**C**). EtOH group (**D**–**F**). (**A**–**D**) in the dorsal hippocampus, moderate intensity expression is observed in the soma of pyramidal neurons in the CA1 sector in the ISO group, while it is minimal in the same region in the EtOH group. (**B**,**E**) in the striatum, in the ISO group, neurons with large and polygonal somas that intensely express Cxcl16, as well as neuropile, are observed, while in the EtOH group the expression is of lesser intensity and in fewer cells. (**C**,**F**) periaqueductal gray substance, the intensity and number of neurons that express Cxcl16 in their soma and in dendritic projections is higher in the ISO group compared to the expression in the same region of the EtOH group. Scale bar = 20 µm.

**Table 1 brainsci-10-00987-t001:** Functional annotation up-and downregulated genes.

Functional Annotation	No. of Genes	Enrichment Score	Benjamini
Transcription regulation	104	6.69	5.9 × 10^−3^
Homeobox	19	2.04	7.2 × 10^−2^
Transcription factor fork head	6	2.00	1.02 × 10^0^
Nuclear hormone receptor	7	1.87	1.0 × 10^0^

Note: the enrichment value a modified form of the *p* value of the exact Fisher test. Benjamini in DAVID requests adjusted *p*-values by using the method of Benjamini and Hochberg.

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
