# Peer review of "Prenatal Alcohol Exposure in Rats Diminishes Postnatal Cxcl16 Chemokine Ligand Brain Expression"

_brainsci, 2020, doi:10.3390/brainsci10120987_

Round 1

Reviewer 1 Report

  • The authors measured gene expression profiles using microarrays, and used real-time PCR for confirmation of high z-score genes. The authors further confirmed the down-regulated expression of chemokine Cxcl16 in multiple brain regions using immunohistochemistry.
  • Their in vivo model of chronic prenatal exposure to EtOH has already shown good results in mimicking the main characteristics of fetal alcohol spectrum disorder in a previous study, so its use in the current study is appropriate.
  • When comparing pups that were prenatally exposed to ethanol (EtOH group) to control pups prenatally exposed to sucrose solution (ISO group), the authors found 945 differentially expressed genes in the Nucleus Accumbens. A majority of the genes decreased in expression - 636 genes as opposed to only 309 genes that increased.
  • However, when the authors chose four genes with high z-scores to confirm their expression changes in the Nucleus Accumbens. Only two out of the four genes were able to be confirmed with real-time PCR, making the reviewer concerned about the validity of the microarray. The authors also showcase PCR expression changes in the prefrontal cortex but the results were either non-significant or did not seem to add much to the story, since the remainder of the paper focuses on Cxcl16 (Figure 2).
  • Focusing on Cxcl16, immunohistochemistry of multiple brain regions, including the Nucleus Accumbens and prefrontal cortex, generally show higher expression of Cxcl16 in the ISO group when compared to the EtOH group, so the conclusion is appropriate.

Author Response

Reviewer # 1

  1. Their in vivo model of chronic prenatal exposure to EtOH has already shown good results in mimicking the main characteristics of fetal alcohol spectrum disorder in a previous study, so its use in the current study is appropriate. Focusing on Cxcl16, immunohistochemistry of multiple brain regions, including the Nucleus Accumbens and prefrontal cortex, generally show higher expression of Cxcl16 in the ISO group when compared to the EtOH group, so the conclusion is appropriate.

RESPONSE:  We thank the reviewer for his/her generally positive comments on our work, both regarding the appropriate use of the model and for the presented conclusions.

  1. However, when the authors chose four genes with high z-scores to confirm their expression changes in the Nucleus Accumbens. Only two out of the four genes were able to be confirmed with real-time PCR, making the reviewer concerned about the validity of the microarray.

RESPONSE: The reviewer makes an important point. Part of the possible explanation is that we found a very variable expression across the different brain regions for some of the genes tested. Consequently, although the dissections of the nuclei accumbens were performed with care, the possibility exists that a little bit more tissue was included from surrounding areas in some of the multiple samples taken for the qPCR confirmation. That is precisely why it was mandatory to confirm each gene expression, in addition to running qPCR, by performing an immunohistochemical analysis, to highlight the expression pattern of a gene of interest. Regarding the four genes of interest, we received commercially available antibodies for two of them: Cxcl16 and Fkbp5. As a result, at this moment we cannot move forward and derive conclusions directly from the whole list of genes found; rather, we need to corroborate the genes of interest one by one.

In attention to the reviewer´s comment, we re-wrote a paragraph in the result section 3.3 qPCR validation of microarray results, to clearly explain the obtained results and the lack of confirmation for two of the selected genes, as follows (lines 204-213):

“For ISO/VH-EtOH/VH, the downregulation of FK506 binding protein 5 (Fkbp5) (p= 0.02) and C-X-C motif chemokine ligand 16 (Cxcl16) (p = 0.04) was confirmed, under these experimental conditions (Figure 1). The high mobility group 20B (Hmg20b) (p = 0.29) was not confirmed. Because there are commercially available antibodies for Cxcl16 and Fkbp5, we further explored their brain expression by immunohistochemistry. For Hmg20b there is no antibody available, so further analysis of its expression is not currently possible. Finally, for the confirmation of homeobox B13 (Hoxb13) by real-time PCR, we obtained a commercially available probe which only corresponded to the mouse gene sequence (the rat sequence was not available). Consequently, we were unable to amplify this gene for verification. Further analysis needs to be undertaken in this case”.        

Reviewer 2 Report

This study investigate the consequence of prenatal alcohol exposure (PAE) on gene expression in the rat nucleus accumbens. Using a RNA microarray approach, the authors highlight 945 differentially expressed genes. Among these genes, authors focus on the expression of C-X-C motif chemokine ligand 16 (Cxcl16).  They show a down-regulation of Cxcl16 in PAE rats and a possible role of chemokine signaling in the pathology.

The manuscript is well written but I’m not convince by the methodology used in the study and as well as the conclusion of the study.

Several major issues should be addressed:

  • In the prenatal administration protocol, the control group received a solution of sucrose (10.5 g/Kg). Sucrose act on reward circuit including nucleus accumbens (for review see Wiss DA et al 2018). Moreover, maternal high-sugar diet influence cocaine-seeking behavior with modification of gene expression in the nucleus accumbens (Gawliński et al 2020). Authors should at least a control that regulation of Cxcl16 mRNA observed in the nucleus accumbens is due to a prenatal alcohol exposure and not due to a prenatal sucrose exposure
  • In this study, authors examined the gene expression profile in the rat nucleus accumbens using microarrays. Unfortunately, no data of the microarray experiment is exposed in the manuscript. Authors has to present the list of differentially expressed genes including, the name of the gene, the refseq accession number, the fold change of regulation, and the Benjamini-Hochberg Adjusted P value.
  • HRP visualization system is non-quantitative method and can’t be used to measured modification of Cxcl16 protein expression in PEA and control rats. Authors should only explored Cxcl16 distribution between the two groups.
  • Recombinant Cxcl16 has a molecular weight of 10.1 kDa. The antibody angainst Cxcl16 used in the study recognize of protein with a molecular weight of 40 kDa (wester-blot; https://www.scbt.com/fr/p/cxcl16-antibody-c-5?requestFrom=search). How authors explain this discrepancy? How did they validate the antibody used in the study?
  • The microarray results ought to be deposited either in the GEO database or in the NCBI’s Sequence Read Archive according to Brain Sciences instructions.

Author Response

Reviewer # 2

  1. In the prenatal administration protocol, the control group received a solution of sucrose (10.5 g/Kg). Sucrose act on reward circuit including nucleus accumbens (for review see Wiss DA et al 2018). Moreover, maternal high-sugar diet influence cocaine-seeking behavior with modification of gene expression in the nucleus accumbens (Gawliński et al 2020). Authors should at least a control that regulation of Cxcl16 mRNA observed in the nucleus accumbens is due to a prenatal alcohol exposure and not due to a prenatal sucrose exposure

RESPONSE: The reviewer brings up a relevant point. There are some additional considerations, however. In the paper of Gawliński et al. (2020) they administered a high sugar diet (HSD), consisting of 70% carbohydrates, of which 44% was sucrose. Laboratory rats usually consume a standard chow diet, composed of nearly 58% carbohydrates of which less than 5% is sugar. Moreover, the administration paradigm of the HSD was prolonged from G1 until PND21 (42 days in total). By contrast, in our study the period of exposure to the isocaloric solution was shorter and restricted to gestation (G8-G20) (13 days in total, less than one third of the other study). The amount of sucrose in the solution is equivalent to the caloric contribution of ethanol and has shown no interference and no significant differences in other similar experimental designs (Wang et al., 2019 [10.1016/j.bbr.2018.07.030]). Additionally, they only evaluate the expression of melanocortin 4 receptor.

On the other hand, as the reviewer indicates, although the relationship between the reward circuit and the orexigenic peptides has been documented in the nucleus accumbens, in numerous studies of prenatal alcohol exposure the isocaloric group and the untreated group show no differences in the levels of expression of these peptides (Chang et al., 2018 [10.1523/JNEUROSCI.0637-18.2018]; Chang et al., 2020 [10.1016/j.neuroscience.2019.10.013]), indicating that extraordinarily high levels of sugar are required for the effect reported by Gawliński´s group.

Nevertheless, and in attention to the reviewer´s interesting comment, we add a paragraph in section 2.2. Prenatal alcohol treatment, to explain the reason why we use an isocaloric group as a control for the alcohol group, and include references to clarify the point raised, as follows (lines 92-97):

This isocaloric group has been used as a control group in previous studies using this model, showing almost no interference (Wang et al., 2019). In addition, although the relationship between the reward circuit and the orexigenic peptides have recently been documented in the nucleus accumbens (Wiss et al., 2018), nevertheless in studies of PAE, the isocaloric group and the untreated group show no differences in the levels of expression of these peptides (Chang et al., 2018; Chang et al., 2020)

If the reviewer considers it indispensable that we perform an immunohistochemistry in the brain of intact animals to demonstrate that there are no differences with the isocaloric group in the expression pattern of Cxcl16, we would ask for more time to perform such  experiments. However, the sacrifice of more animals is probably unnecessary because the objective of this work is to observe the effects of alcohol and not the effects of sugar. The rationale of the control group as an isocaloric group is precisely to have a control to avoid the effect of light over-calories administration for the alcohol.

At this point it is also relevant to mention that the protocol here realized was previously registered and received the approbation of an Ethics Committee (the original letter from the committee has been sent to the Brain Sciences journal editorial office). Therefore, we include the following sentence in the section 2.1 (lines 78-80):

"This project was approved by the Ethical Committee of the Instituto de Neurociencias in the Centro Universitario de Ciencias Biológicas y Agropecuarias, Universidad de Guadalajara (ET-112016-225)".

  1. In this study, authors examined the gene expression profile in the rat nucleus accumbens using microarrays. Unfortunately, no data of the microarray experiment is exposed in the manuscript. Authors has to present the list of differentially expressed genes including, the name of the gene, the refseq accession number, the fold change of regulation, and the Benjamini-Hochberg Adjusted P value.

RESPONSE: In attention to the reviewer request, we incorporate as supplementary tables, the list of genes modified derived from the microarray (up and down). We want to make clear that the whole and deep analysis of the performed microarrays would be the subject of a different publication. In the present work we only focused on the expression of Cxcl16 and include the mention of the experiments that led us to identify this particular gene.

Consequently, we added two supplementary tables, cited in the section 3.1. Changes in patterns of gene expression in the nucleus accumbens after PAE, as follows (line 188-190): “Complete lists of genes are included in supplementary tables S1 and S2.”

Additionally, we have also added the correspondent Benjamini Hochberg value for each gene cluster in the Table 1 (lines 198-199, and Table 1).

  • Table S1 shows the 636 differentially expressed genes with a z-score of ≤ 2.00 after prenatal alcohol exposure including the accession number and the gen symbol
  • Table S2 shows the 309 differentially expressed genes with a z-score of ≥ 2.00 after prenatal alcohol exposure including the accession number and the gen symbol
  1. HRP visualization system is non-quantitative method and can’t be used to measured modification of Cxcl16 protein expression in PEA and control rats. Authors should only explored Cxcl16 distribution between the two groups.

RESPONSE:  We thank the reviewer for pointing this out. Although it is not a quantitative method, the HRP-based method of immunostaining in paraffined tissue led to the identification of higher or lower number of cells that express this cytokine in different brain regions. The histological analysis performed considers several aspects, such as anatomical region of expression, extension of expression within the same region, different cell morphologies that expresses the protein, sub-cellular region (nucleus, cytoplasm, cytoplasmic membrane) and intensity of expression, in both study groups. Although it is possible to make automated measurements of the intensity of expression, this method does not distinguish between the different variables mentioned, which makes it necessary for an expert to interpret the stain. Thus, the results of this descriptive analysis allow us to contrast, in correlation with the results of previous experiments, the expression of Cxcl16 in a specific way, which contributes to the understanding of the dynamics of expression of this cytokine in the tested model.

  1. Recombinant Cxcl16 has a molecular weight of 10.1 kDa. The antibody angainst Cxcl16 used in the study recognize of protein with a molecular weight of 40 kDa (wester-blot; https://www.scbt.com/fr/p/cxcl16-antibody-c-5?requestFrom=search). How authors explain this discrepancy? How did they validate the antibody used in the study?

RESPONSE: We appreciate the opportunity to explain this aspect. As the reviewer correctly  pointed out, the expected molecular weight of a human recombinant CXCL16 theoretically is 10.2 kDa. Nevertheless, the observed molecular weight of the protein may vary from the predicted molecular weight, as is indicated in the Disclaimer note from the producer: “The observed molecular weight of the protein may vary from the listed predicted molecular weight due to post translational modifications, post translation cleavages, relative charges, and other experimental factors” (https://www.novusbio.com/products/recombinant-human-cxcl16-protein_976-cx). It is also worth considering that the native forms of this protein in humans are 14, 28 and 50 kDa (Tohyma et al., 2007). Anti-Cxcl16 is a mouse monoclonal antibody from Santacruz recommended for detection of CXCL16 of human origin, but this is an ortholog gene with a high degree of conservation among humans and rodents. In the present study we worked with the rat Cxcl16 in rat tissue in its native form, whose predicted molecular weight is 27.2 kDa

Validation for the Cxcl16 antibody used was performed by staining control tissues, from both human and rat, previously reported to constitutively express this particular cytokine. A 1:50 dilution was used, as recommended by the manufacturer in immunocytochemistry application. In Supplementary Figure 1 we show some of these results that validate the Cxcl16 staining.

We add this explanation to section 2.8. Immunohistochemistry, as follows (line 174-177):

“For validation and standardization of Cxcl16 antibody staining, we performed immunohistochemistry for control tissues, both from human and rat, previously reported to constitutively express this cytokine. Results are presented in Supplementary Figure 1”.

  1. The microarray results ought to be deposited either in the GEO database or in the NCBI’s Sequence Read Archive according to Brain Sciences instructions.

RESPONSE: We thank the reviewer for bringing this aspect to our attention. Consequently, we entered the complete results from the microarrays performed in GEO public database. We indicate this in the results section as follows (lines 188-190): “Obtained results from microarray hybridization were entered in GEO database under the accession number: GSE160433”

Finally, we would like to thank both reviewers for their detailed revision of our manuscript and for their constructive and highly appreciated comments and suggestions, which have been extremely helpful in improving the manuscript.

Round 2

Reviewer 2 Report

The authors were not able to address my concern:

  • The Benjamini-Hochberg Adjusted P values has to perform done on the raw data (z-score) not on the PAE analysis. Authors to has to: a) calculated a FDR on the microarray data and b) performed the PAE analysis only with the genes with a p<0.05
  • The microarray data deposit in the GEO database is not supporting MIAME-compliant (https://www.ncbi.nlm.nih.gov/geo/info/MIAME.html; Brazma A et al 2001)
  • Authors admit that the HRP visualization system is “not a quantitative method” but no modification of the manuscript have been made. The manuscript always refers to a quantitative analysis of the immunohistochemistry For example, line 241  “…cytoplasmic staining of moderate intensity”,  line 243 “…cytoplasmic staining of moderate intensity.”, line 250 “…from moderate to high intensity”, line 251 “Under PAE, by contrast, the intensity of the expression was slight”
  • The supplementary experiment presented in figure S1 is a way to validate the secondary antibody but not the specificity of the antibody targeting CXCL16. To validate the primary antibody, authors should at least performed immunohistochemistry experiments in presence of saturated concentration of CXCL16
  • The new bibliographic references added to the manuscript are not referenced.

Author Response

Answers to Reviewer 2

Comment 1- The Benjamini-Hochberg Adjusted P values has to perform done on the raw data (z-score) not on the PAE analysis. Authors to has to: a) calculated a FDR on the microarray data and b) performed the PAE analysis only with the genes with a p<0.05

Response: We thank the reviewer for pointing this out. The Microarray experiments were performed as previously reported in a considerable number of works by our group (a couple of them are mentioned here for reference). The microarray results were analyzed using genArise software (http://www.ifc.unam.mx/genarise/), as indicated in the material and methods section. genArise is a completely validated package (see mentioned references) that contains specific functions with which to perform an analysis of cDNA microarrays data to detect genes that are significantly differentially expressed under different conditions. Before this analysis, genArise carried out several transformations on the data to eliminate low-quality measurements and to adjust the measured intensities in order to facilitate comparisons.

The selection of the differentially expressed genes is achieved by calculating an intensity-dependent Z-score. genArise uses a sliding window algorithm to calculate the mean and standard deviation (SD) within a window surrounding each data point and define a Z-score.

With these criteria, we believed the elements with (Z-score) ≥ 1.5 SD would be the significantly expressed genes. To be more rigorous, only those genes with ≥ 2.0 SD were considered before proceeding with the DAVID analysis. Therefore, as we understand, these differentially expressed genes correspond to the genes that the reviewer indicates must be included to perform the analysis (only those with a statistical probability to be significant). Consequently, to better explain this aspect indicated by the reviewer, we include the following brief explanation in the material and methods section (lines 129-133):

“According to the genArise analysis, the genes with a z-score +/- 1.5 are considered  statistically significant in difference in expression (p<0.05). However, to be more rigorous, we select only those genes with z-score over +/- 2.0 (a higher selective z-score), to be included for further analysis in the DAVID software (see the next section).”   

Comment 2- The microarray data deposit in the GEO database is not supporting MIAME-compliant (https://www.ncbi.nlm.nih.gov/geo/info/MIAME.html; Brazma A et al 2001)

Response: We carefully review the six main points highlighted in the MIAME guidelines. Moreover, according to the Gene Expression Omnibus website: “All GEO submission procedures are designed to closely follow the MIAME and MINSEQE checklists; if you provide all requested information, your submission will be compliant”. (https://www.ncbi.nlm.nih.gov/geo/info/MIAME.html).

All the information requested can be found with the accession number GSE160433:

https://www.ncbi.nlm.nih.gov/geo/query/acc.cgi?acc=GSE160433

Information about the platform can be found as a part of our submission at: https://www.ncbi.nlm.nih.gov/geo/query/acc.cgi?acc=GPL10521

Information about the sample can be found as a part of our submission at: https://www.ncbi.nlm.nih.gov/geo/query/acc.cgi?acc=GSM4873326

Relating to this reviewer’s observation, we add a brief sentence to section 2.5 of material and methods (line 133-136):

“The microarray data were deposited in the GEO database, in line with the MIAME (Minimum Information About a Microarray Experiment) and MINSEQE (Minimum Information About a Next-generation Sequencing Experiment) guidelines.”

Comment 3- Authors admit that the HRP visualization system is “not a quantitative method” but no modification of the manuscript have been made. The manuscript always refers to a quantitative analysis of the immunohistochemistry For example, line 241  “…cytoplasmic staining of moderate intensity”,  line 243 “…cytoplasmic staining of moderate intensity.”, line 250 “…from moderate to high intensity”, line 251 “Under PAE, by contrast, the intensity of the expression was slight”

Response: Perhaps our initial explanation in this respect was not clear enough. The reviewer notes that “HRP visualization system is not a quantitative method”, and we agree. Our objectives with this experiment were: 1) to validate the expression of the protein in the brain regions where Cxcl16 gene expression was demonstrated; 2) to document the type and magnitude of cells that expressed this protein, and 3) to qualitatively assess the intensity of CxCl16 expression. We are aware that although this methodology is not a precise quantitative method per se (unless an optic quantification of the labeling is performed by software), it is useful enough to highlight more or less intensity of labeling (not how much more or less in specific amounts or percentages), as well as the number of labeled cells, being described and qualified by an anatomic pathology expert. We know that other methods such as western blot or ELISA are necessary to make quantitative or semiquantitative comparisons of the protein expression, however, that was not our aim in this work. Therefore, the terms we used in the description of the observed labeling for Cxcl16 in different brain regions ("moderate", "high" or "slight" intensity), are not quantitative terms, but comparative ones. It does not indicate specific amounts (quantities) but instead indicates evident qualitative differences between treated and control (isocaloric) animals.

To attend to the reviewer’s comment, and to emphasize the qualitative nature of the observations described, we add the following text in section 3.4 of the results (line 245-247 and 255):

Line 245-247: “…exhibited cytoplasmic staining of apparently moderate intensity. In the EtOH group, the somas and neuropile of the shell region also showed a suggestively moderate staining (Figure 3). Quantitative methods are required to complement the results to confirm our observations”.

Line 255: “…ranged from moderate to high intensity varying between the different layers…”

Comment 4- The supplementary experiment presented in figure S1 is a way to validate the secondary antibody but not the specificity of the antibody targeting CXCL16. To validate the primary antibody, authors should at least performed immunohistochemistry experiments in presence of saturated concentration of CXCL16.

Response: In the supplementary figure 1, two aspects are presented: the specificity of the primary antibody and the validation of the secondary antibody. In our study, validation was performed with rat and human tissues in which Cxcl16 is constitutively expressed such as lung, adrenal gland and testicle (based on human protein atlas: https://www.proteinatlas.org/humanproteome/tissue, and Uniprot: https://www.uniprot.org/). These tissues are considered positive external anatomical controls, in which the expression of the antigen under basal conditions is known a priori. The expected negativity in structures such as the cell nucleus functions as a negative internal control that helps us to rule out a non-specific interaction of the primary antibody with epitopes other than the predicted target. Therefore, with these controls, we gather with a high degree of certainty that the antibody we used indeed identifies Cxcl16 in tissue samples, at a 1:50 dilution, as recommended by manufacturer in immunocytochemistry application. Finally, the tests for the specificity of the antibody must be performed by each manufacturer before they release the product to commercialization, or when the antibody is home-made by the researchers. In the case of this antibody, the manufacturer presents evidence for the specific identification of the protein by western blot and after the expected electrophoretic migration (https://www.scbt.com/p/cxcl16-antibody-c-5).

Furthermore, as the reviewer indicates, figure S1 also validates the secondary antibody, comparing panels A, B and C with its correspondent controls in panels D, E and F. The negativity of the stain in the absence of the primary antibody allows us to rule out a false positive label due to non-specific binding of the secondary antibody. To support our observations on tissue expression of Cxcl16, we must emphasize that all tissues analyzed by immunohistochemistry were subjected to the same pre-analytical and analytical conditions. We consider that the control of these variables confers validity to the interpretation of the results.

To address this reviewer's observation, and for better clarification in the manuscript, we have adjusted a previously edited section of the text (section 2.8, lines 179-182):

“For validation and standardization of Cxcl16 antibody staining, we performed immunohistochemistry in positive external anatomical controls tissues with known constitutive Cxcl16 expression (both from human and rat). Results are presented in Supplementary Figure 1.”

Comment 5- The new bibliographic references added to the manuscript are not referenced.

Response: Thank you very much for this observation, we had missed one reference. We double checked and now all references are properly added to the reference list.
